# Lipid Profile Rather Than the *LCAT* Mutation Explains Renal Disease in Familial LCAT Deficiency

**DOI:** 10.3390/jcm8111860

**Published:** 2019-11-03

**Authors:** Itziar Lamiquiz-Moneo, Fernando Civeira, Diego Gómez-Coronado, Francisco Blanco-Vaca, Hilda Mercedes Villafuerte-Ledesma, Miriam Gil, Nuria Amigó, Rocío Mateo-Gallego, Ana Cenarro

**Affiliations:** 1Lipid Unit, Hospital Universitario Miguel Servet, Instituto de Investigación Sanitaria Aragón (IIS Aragón), CIBER Cardiovascular (CIBERCV), 50009 Zaragoza, Spain; itziarlamiquiz@gmail.com (I.L.-M.); rmateo@unizar.es (R.M.-G.); ana.cenarro@gmail.com (A.C.); 2Department of Internal Medicine, Psychiatry and Dermatology, Universidad de Zaragoza, 50009 Zaragoza, Spain; 3Department of Biochemistry-Research, Hospital Universitario Ramón y Cajal, IRYCIS, CIBER de Fisiopatología de la Obesidad y Nutrición (CIBEROBN), 28034 Madrid, Spain; diego.gomez@hrc.es; 4Hospital de la Santa Creu i Sant Pau, Servicio de Bioquímica—IIB Sant Pau, CIBER de Diabetes y Enfermedades Metabólicas Asociadas (CIBERDEM), Departament of Biochemistry and Molecular Biology, Universidad Autónoma de Barcelona, 08041 Barcelona, Spain; FBlancoVa@santpau.cat; 5Nephrology Department, Hospital Clínico Universitario Lozano Blesa, IIS Aragón, 50009 Zaragoza, Spain; hilda.villafuerte21@gmail.com; 6Biosfer Teslab, SL, 43201 Reus, Spain; mgil@biosferteslab.com (M.G.); nuriaamigo@gmail.com (N.A.); 7Metabolomics Platform, Universidad Rovira i Virgili (URV), Instituto de Investigación Sanitaria Pere Virigili (IISPV), 43003 Tarragona, Spain; 8CIBER de diabetes y enfermedades metabólicas asociadas (CIBERDEM), 28029 Madrid, Spain; 9Department of Physiatry and Nursing, Universidad de Zaragoza, 50009 Zaragoza, Spain

**Keywords:** familial LCAT deficiency, fish eye disease, VLDL, NMR, FPLC, triglycerides, phospholipids

## Abstract

Renal complications are the major cause of morbidity and mortality in patients with familial lecithin–cholesterol acyltransferase (LCAT) deficiency (FLD). We report three FLD patients, two of them siblings—only one of whom developed renal disease—and the third case being a young man with early renal disease. The aim of this study was to analyze the clinical characteristics and possible mechanisms associated with renal disease in these patients. Plasma lipid levels, LCAT activity, lipoprotein particle profile by NMR and FPLC, free and esterified cholesterol, presence of lipoprotein X (LpX) and DNA sequencing in the three FLD patients have been determined. The three cases presented clinical characteristics of FLD, although only one of the siblings developed renal disease, at 45 years of age, while the other patient developed the disease in his youth. Genetic analysis revealed new missense homozygous mutations, p.(Ile202Thr) in both siblings and p.(Arg171Glu) in the other patient. Lipoprotein particle analysis showed that the two patients with renal disease presented higher numbers of small very low-density lipoprotein (VLDL) and a higher concentration of triglycerides in VLDL. This study reports three new cases of LCAT deficiency, not previously described. Renal disease is not only dependent on LCAT deficiency, and could be due to the presence of VLDL particles, which are rich in triglycerides, free cholesterol and LpX.

## 1. Introduction

Lecithin cholesterol acyltransferase (LCAT) is a lipoprotein-associated enzyme, which plays a central role in the esterification of free cholesterol (FC) in plasma, in the formation of mature high density-lipoprotein (HDL) particles, and in the intravascular stage of reverse cholesterol transport [1,2]. Human LCAT is a glycoprotein of 416 amino acids, encoded by a gene located on chromosome 16q21-22, which esterifies the majority of the free cholesterol located at the surface of lipoprotein particles, by catalyzing the translocation of fatty acid moiety in sn-2 position of lecithin to the free 3-OH group of cholesterol [3].

LCAT deficiency is a rare autosomal recessive disease, with prevalence below 1:1,000,000. Nearly 100 different mutations in the *LCAT* gene have been identified so far [4]. Mutations in both alleles of LCAT gene may lead to two syndromes, namely familial LCAT deficiency (FLD) and fish eye disease (FED). Both syndromes share several abnormalities in lipoproteins, including a decrease in HDL cholesterol level, apo A-I and apo A-II, a decrease in LDL cholesterol level and an increase in FC and apo E [2,5]. In FLD, the LCAT enzyme is either absent in plasma, or exhibits no catalytic activity on any lipoprotein [6]. However, FED is characterized by a partial enzyme deficiency, in which LCAT does not esterify cholesterol in HDL, but retains some activity on lipoproteins containing apo B (VLDL and LDL) [7]. Individuals with FED have very low levels of HDL cholesterol and corneal opacities [3,8]. In FLD, free cholesterol accumulates in all plasma lipoproteins, and patients present HDL cholesterol deficiency, corneal opacification, hemolytic anemia, hypertension, hypertriglyceridemia and proteinuria, frequently progressing to end stage renal disease [5].

Renal involvement is the major cause of morbidity and mortality in FLD patients. It usually starts as proteinuria and progresses to renal disease. The cause of renal disease in FLD is not well understood, but has been attributed to the deposition of FC-rich lipoproteins, called Lipoprotein X (LpX), without hydrophobic core lipids particles, in the glomeruli and renal arterioles [9,10]. FC deposits appear as small vacuolations of various sizes on the epimembranous, intramembranous, and subendothelial spaces of the glomerular basement membrane and in the mesangium [5]. These deposits are suggested to be the main mechanism behind the development of renal disease [2,11].

We have had the opportunity to study three patients with FLD, two of them siblings sharing the same LCAT mutation. Interestingly, despite the identical genotype present in both siblings, only one developed renal disease. The third case studied was a young man with early renal disease. The aim of this study was to analyze the clinical characteristics of, and possible mechanisms associated with, the phenotype differences in these patients.

## 2. Materials and Methods

### 2.1. Subjects

We present three cases, two of them siblings, with LCAT deficiency, attending our Lipid Unit, who were referred for extremely low levels of HDL cholesterol. The Lipid Unit is located in the Hospital Universitario Miguel Servet, which is the reference center for disorders of lipid metabolism in the Aragon region in Spain, which has approximately 1,300,000 habitants.

### 2.2. Lipid Analysis

Lipid and lipoprotein analyses were performed on Ethylenediaminetetraacetic acid (EDTA) plasma samples collected after at least a 10 h overnight fast, without intake of lipid lowering drugs in at least 6 weeks. Total cholesterol and triglycerides (TG) levels were determined by standard enzymatic methods. High density lipoprotein cholesterol (HDL cholesterol) was measured by an enzymatic reaction using cholesterol oxidase (UniCel DxC 800; Beckman Coulter, Inc, Brea, CA). Lipoprotein(a), apo A-I, apo B, and C-reactive protein were determined by IMMAGE kinetic nephelometry (Beckman-Coulter, Inc). LDL cholesterol was calculated by Friedewald’s formula. All subjects signed an informed consent form, to a protocol previously approved by our local ethics committee (Comité Ético de Investigación Clínica of Aragon, PT13/0010/0017, Zaragoza).

### 2.3. Plasma LCAT Activity

Plasma LCAT activity was determined using reassembled HDL, containing 1-palmitoyl-2-oleyl-sn-3-phosphocholine, [^3^H] cholesterol, and apo A-I. Endogenous LCAT activity was measured using the lipoproteins of whole and apo B-depleted plasma as substrate, and expressed as fractional esterification rate (FER) [12].

### 2.4. Genetic Analysis

Whole blood genomic DNA was isolated using standard methods. Promoters, coding regions, and intron-exon boundaries of *LCAT* (NM_012108.3) were amplified by polymerase chain reaction and purified by ExoSap-IT (USB). Amplified fragments were sequenced by Sanger method, using the BigDye 3.1 sequencing kit (Applied Biosystems) in an automated ABI 3500xL sequencer (Applied Biosystems). DNA sequences were analyzed using Variant Reporter software (Applied Biosystems). The genetic analysis was extended to all available family members of the siblings (Appendix A).

### 2.5. Lipoprotein Particle Analysis 

Lipoprotein particle concentrations were measured by NMR, based on the LipoScale test ^®^. The lipid concentration (i.e., triglycerides and cholesterol) and the mean size of VLDL, LDL and HDL, as well as the particle number of nine subclasses (large, medium and small VLDL, LDL and HDL) were determined as previously reported [13]. Briefly, particle concentration and diffusion coefficients were obtained from the measured amplitudes, as well as attenuation of their spectroscopically distinct lipid methyl group NMR signals, using the 2D diffusion-ordered 1H NMR spectrometry (DSTE) pulse. The methyl signal was surface-fitted with nine lorentzian functions associated with each lipoprotein subclass—large, medium and small. The area of each lorentzian function was related to the lipid concentration of each lipoprotein subclass, and the size was calculated from their diffusion coefficient.

Each subclass particle concentration was calculated by dividing the lipid volume by the particle volume of a given class. Lipid volumes were determined by using common conversion factors to convert concentration units into volume units. The weighted average of VLDL, LDL and HDL particle sizes were calculated from various subclass concentrations by summing the known diameter of each subclass, multiplied by its relative percentage of subclass particle number.

To assess similarities and differences among LCAT deficiency patients in terms of their lipoprotein profile, we used a previously analyzed reference population from Biosfer Teslab data base, including 6000 subjects from the general population, of men and women of different ages (between 15 and 85 years old), and comparing the study cases with subjects with a normal profile of NMR lipoprotein and low HDL cholesterol (<35 mg/dL). Firstly, we applied a Principal Component Analysis (PCA) model, using the lipoprotein composition, size and the subclass’s particle number as input variables. Then, we applied a Partial Least Square Discriminant Analysis (PLS-DA) model (see below) to optimize the separation between LCAT deficiency patients and the general population, after a variable selection process, by using genetic algorithm methodologies [14].

### 2.6. Gel Filtration Fast Protein Liquid Chromatography (FPLC)

Two hundred and fifty μl of fresh plasma were injected into a Superose 6 HR 10/30 column (Pharmacia) equilibrated with TBS containing 0.02% NaN_3_. Chromatography was performed at a flow rate of 0.3 mL/min and, after elution of the first 5 mL, fractions of 0.5 mL were collected. Cholesterol, triglyceride and choline-containing phospholipid concentrations were measured in the collected fractions by enzymatic methods. A portion of selected fractions was concentrated three-fold by using appropriate centrifugal concentrators (MW cut off 10000 Ultrafree-MC, Millipore). Equal aliquots of the concentrates were subjected to agarose gel electrophoresis and staining for lipids or, alternatively, anti-apo A-I or anti-apo A-II immunoblotting.

### 2.7. LpX Determination

Presence of LpX was analyzed as reported by Freeman et al [15]. Briefly, 10 μL of fresh EDTA-plasma (not frozen) or 10 μL synthetic LpX were loaded on Sebia Hydragel 7 lipoprotein(e) gels (#4134; Sebia, Inc., Norcross, GA). Syntethic LpX was prepared by mixing 6.1 mg L-α-lecithin (Calbiochem, Millipore) with 1.06 mg cholesterol (Sigma–Aldrich, Spain) from their stock solutions in chloroform, and then drying the lipid mixtures under nitrogen. Two mL of normal saline were added, and the mixture was vortexed for 10 min and sonicated for 10 min to generate multilamellar particles enriched in free cholesterol and phospholipid, in the size range of LpX. Sequential dilutions of LpX were loaded onto the gel. Electrophoresis was for 30 min at 100 V in a Hydrasys 2 System (Sebia). Gels were fixed with 10% trichloroacetic acid and incubated with Filipin Stain to detect free cholesterol. Filipin Stain was prepared fresh each time by dissolving 12.5 mg Filipin (Sigma–Aldrich, Spain) in 0.5 mL N, N’ dimethyl formamide, and then adding 50 mL 1XPBS containing 0.1% sodium azide and mixing thoroughly. Gels were incubated overnight in 50 mL Filipin Stain at 4 °C with gentle rocking, shielded from light, and then washed for 20 min at room temp with 50 mL 1XPBS. Imaging was carried out with a UV transilluminator.

We estimated LpX concentration through the measurement of total and free cholesterol. Lipids were extracted from five serum samples (three controls and two LCAT deficiency patients) using a slightly modified BUME method [16], replacing heptane with diisopropyl ether. Dried lipid extracts were reconstituted with CDCl_3_-CD_3_OD-D_2_O (16:7:1), containing 1.18 mM tetramethylsilane (TMS) and transferred to 5 mm NMR glass tubes for analysis. One dimensional ^1^H-NMR spectra were obtained, using a 90° pulse with pre-saturation sequence (zgpr). Spectra were phased, baseline corrected, and referenced (TMS 0 ppm), and then selected peaks (free and esterified cholesterol) were deconvoluted using LipSpin software [17].

### 2.8. Statistical Analysis

Multivariate statistical analyses were computed in MATLAB (Eigenvector Research Inc., Manson, WA, USA). PLS-DA was used as a supervised classification method for LCAT deficiency identification among (i) a general population and ii) individuals with low HDL cholesterol levels. PLS-DA relates the X matrix (experimental data) and the Y matrix (classes of samples), in order to find the maximum discrimination between classes and the maximum covariance between the X and Y matrices simultaneously [18].

## 3. Results

The first patient was referred to our Lipid Unit for an extremely low plasma concentration of HDL cholesterol, corneal opacities and progressive renal disease when he was 65 years old. He reported low plasma levels of total cholesterol, HDL cholesterol, moderately elevated TG and proteinuria since his mid-forties. His lipid values at that time are reported in Table 1. He developed progressive renal disease requiring hemodialysis at the age of 75 and he died 12 years later. The second patient, sister of the first patient, was a 78-year-old female, that was also referred to our Lipid Unit for an extremely low plasma concentration of HDL cholesterol in a routine analysis. This patient reported that she had suffered from corneal opacity during the last 30 years. Currently, she is 84 years old, asymptomatic, and her renal function remains within normal values, including urinalysis without proteinuria. Physical examination showed corneal opacity (Figure 1A). Her blood lipid values confirmed severe hypoalphalipoproteinemia (Table 1). The third patient was a 38-year-old male who had suffered from corneal opacity, low levels of HDL cholesterol and abnormal results of urinalysis from the age of 20. Physical examination showed corneal opacity (Figure 1B). Urinalysis revealed proteinuria, and blood analysis showed mild impaired renal function. The renal biopsy specimen revealed a thickening of capillary walls and the presence of foamy lipid deposits at the capillary and mesangial levels (Figure 2).The cholesterol esterification rate in serum, as a measure of endogenous LCAT activity, in all cases, was nearly absent, as compared with control values (1.70 nmol/h*mL, 0.05 nmol/h*mL and 1.35 nmol/h*mL, respectively). All patients reported null endogenous LCAT activity, with an FER of 1.40%, 1.5% and 1.9% per hour, respectively, below the minimum reference value (value reference: >3.5% by hour) (Table 1).

Genetic analysis of both siblings revealed a homozygous mutation c.605T>C, which is predicted to cause a substitution of isoleucine by threonine at amino acid 202 of LCAT, p.(Ile202Thr). Genetic analysis in the third case showed a homozygous mutation c.512G>A, which is predicted to cause a substitution of arginine by glutamic acid at amino acid 171 of LCAT, p.(Arg171Glu). Both mutations have been classified as pathogenic by bioinformatic analysis (Polyphen-2 [19], MutationTaster [20] and Predict SNP [21]). The first mutation, p.(Ile202Thr), has been previously associated with hypoalphalipoproteinemia in the heterozygosity state by our group [22] (Table 1).

Plasma agarose gel electrophoresis revealed several disturbances in the lipid profile of the patients, in comparison with the other members of their family. As shown in Appendix A, both siblings were deficient in α-migrating lipoproteins. Moreover, pre-β-migrating particles were virtually absent. An accumulation of lipids, with a mobility intermediate between those of β- and pre-β-migrating particles, appeared in the first patient. As regards the plasma β-migrating particles, those from the first and second case showed a slightly slower mobility than in non-affected family members (Appendix A).

Lipid composition of plasma lipoproteins was analyzed by FPLC, as well as the subsequent determination of lipid levels in eluted fractions (Figure 3). As expected, HDL contained scarce cholesterol, and only substantial amounts of phospholipids were detected in the corresponding fractions in the first and second cases. Moreover, HDL phospholipids reached their maximum concentration later, as compared with those from the control, suggesting the predominance of smaller HDL particles in the two patients. Although VLDL levels were higher in the first case than in his sister, both subjects had phospholipid and cholesterol-rich VLDL levels that were higher than the control’s. In addition, both siblings had lower LDL-cholesterol levels, but higher LDL-triglyceride levels, than the control (Figure 3). Agarose gel electrophoresis and immunoblotting for apo A-I and apo A-II of HDL-containing FPLC fractions showed a substantial reduction in the levels of both apolipoproteins in the two patients (Figure 3).

The NMR-Liposcale test, performed in serum from the second and third patients, revealed a distinctive lipoprotein profile, with null value of small HDL particles, under the limit of detection (<0.01µmol/L) and 0.01 µmol/L, respectively, compared with the control population (average small HDL-P = 21.52 ± 5.24 µmol/L), without blatant differences in medium or large HDL particles. Both patients presented higher values of VLDL (large, medium and small), especially in the third case, compared with the control population (Table 2). Figure 4a shows the score plot of the first PLS-DA model, considering the first two latent variables (LVs), designed to discriminate patients with LCAT deficiency among the general population. We applied a second PLS-DA model to discriminate between patients with LCAT deficiency and patients with low HDL cholesterol levels (HDL cholesterol <35 mg/dL), which represent 2.5% of the studied population (Figure 4b). The PLS-DA score plot showed a clear separation between patients with LCAT deficiency and the rest of the population, excepting one individual, whose position in the variance plot was extreme. This individual was later diagnosed with extreme HDL deficiency, and his lipoprotein pattern was even more severe, since it did not present small HDL particles, neither medium nor large HDL particles. 

The study of the distribution of free cholesterol by agarose gel electrophoresis showed a large amount of free cholesterol in Patient 3, and the presence of particles migrating toward the cathode with reverse electrophoretic mobility compared to other lipoproteins, which correspond to LpX particle (Figure 5).

^1^H-NMR analysis revealed an abnormal lipid profile in the serum of LCAT deficiency patients, compared to controls (Table 3). As expected, mean cholesterol composition in control serum was 28% free and 72% esterified. In contrast, serum from LCAT deficiency patients was highly enriched in free cholesterol. Cholesterol composition in the second case was 67% free and 33% esterified, suggesting the presence of some LpX. Interestingly, the third case had 98% free and 2% esterified cholesterol composition, confirming the presence of high levels of LpX. Appendix A shows the ^1^H-NMR spectra of serum lipids, highlighting the signals of free and esterified cholesterol.

## 4. Discussion

In the present study, we report three cases with FLD with corneal opacities, due to two homozygous mutations (c.605T>C, c.512G>A) not previously described in patients with LCAT deficiency. The mutation, c.605T>C, which produces a substitution of isoleucine by threonine at amino acid 202 of LCAT, p.(Ile202Thr), had been previously described by our group in heterozygosity, in patients with hypoalphalipoproteinemia [22]. The mutation c.512 G>A, which is predicted to cause the substitution of arginine with glutamic acid at amino acid 171 of LCAT, p.(Arg171Glu), has not been previously reported, although Taramelli et al. described another mutation at the same position, which produces a substitution of arginine with threonine at the same amino, and also causes LCAT deficiency [6]. The current study shows that the main cause of mortality in LCAT deficiency does not depend only on *LCAT* mutation, because two of our cases are siblings. Despite sharing the same mutation, just one of them developed renal disease.

LCAT adopts an α/β hydrolase fold, with a predominately parallel β-sheet sandwiched on both sides by α-helices [23]. The mutation p.(Arg171Glu) is located close to Ser181, which seems to be responsible, together with Cys31 and Leu182, for the possible oxyanion hole amino acid residues. The mutation p.(Ile202Thr) is located close to the second subdomain, formed by the insertions between β6/β7 and β7/αE, which is in much closer proximity to the catalytic site [23,24]. Our data are in agreement with the fact that previously reported *LCAT* mutations causing FED are located in the same domains of *LCAT* gene as those causing FLD, such as, for example, the mutations p.(Pro34Gln) [25] and p.(Pro34Leu) [26], reported in FED and FLD, respectively, although both are located in the same amino acid and in close proximity to the oxyanion hole amino acid 31 [23].

One of the most striking discoveries of our study, when analyzing the differences between the lipoprotein distribution by different methods, between the two FLD patients with renal disease and the FLD patient without renal disease, was the presence of abnormal lipoprotein profile in FPLC with higher concentrations of small–medium VLDL in NMR. Similar data have been reported by other studies [7,27]. LpX was confirmed by agarose gel electrophoresis and filipin-staining, and estimated by the distribution of FC and CE by NMR. Some studies reported that LpX is a nephrotoxic particle, that induces all the histological and functional renal hallmarks of FLD [9,10]. Besides, Vaisman et al. have reported on the close relationship between LCAT and LpX, as intravenous treatment with recombinant LCAT in mice^lcat-/-^ restored the normal lipoprotein profile, eliminated LpX in plasma and markedly decreased proteinuria [15]. 

It seems obvious that the *LCAT* genotype, which mostly determines the activity of the LCAT enzyme, is not uniquely responsible for renal disease in patients. Differences between FLD and FED are based on the cholesterol esterification in VLDL [8]. It is also evident that, in cases of LCAT deficiency, the complete metabolism of lipoproteins would be altered, presenting larger VLDL with a high concentration of free cholesterol and of phospholipids in plasma and glomeruli, which all may be factors in the deterioration of kidneys in patients with a familial LCAT deficiency [28]. We have analyzed different biochemical and clinical parameters and we have found that the main differences do not reside in the HDL particle. FPLC and NMR detect both free and esterified cholesterol; besides, FPLC has been previously used in patients with biliary cirrhosis, who showed an abnormal distribution pattern of lipoproteins, with a large accumulation of phospholipids and LpX [29]. In the current study, we confirm that the difference between subjects with renal disease resides in medium size VLDL with high triglyceride and cholesterol content, and the presence of LpX. To date, renal disease has been attributed to the cholesterol esterification of lipoproteins containing apolipoprotein B (VLDL and LDL) [7,8], but we report here two siblings with LCAT deficiency, who share the same LCAT mutation and negligible LCAT activity, only one of whom developed renal disease, which suggests that other genetic or environmental factors can influence the development of renal disease. Hence, the presence of renal disease could be explained by gene–gene or environment–gene interactions, that would favor the presence and amount of LpX. According with this, Rial-Crestelo D et al. [30] reported the case of one family, in which one member was diagnosed as FLD, while their sister was found to have FED, carrying the same LCAT mutation. However, the FED subject was too young to discard the possibility of future renal disease.

In our study, FLD patients with renal disease showed higher numbers of small and medium VLDL particles, and higher concentrations of triglyceride and cholesterol in VLDL by FPLC and by NMR, than in the patient who has not developed renal disease. The substantial concentration of phospholipids in the HDL fraction by FPLC is consistent with LCAT deficiency, as previously described [7]. Advanced NMR testing determined no presence of small HDL particles in the serum of LCAT deficiency samples. Nascent preβ-HDL particles were not detected by the Liposcale test, as these particles do not carry either triglycerides or cholesterol within their core. Therefore, levels of small HDL particles determined by Liposcale were below the limit of detection of the NMR-derived particle size. The weighted HDL particle size is calculated from the subclass particle number multiplied by its associated diameter. In cases of LCAT deficiency, small HDL particles are not detected, and, therefore, size distribution is shifted towards larger particles. However, comparing NMR data from LCAT deficiency patients and subjects with low HDL cholesterol levels, there were differences in some lipoprotein variables, which identified a distinctive lipoprotein profile in patients with LCAT deficiency. FLD patients showed an increased particle-weighted HDL mean size (>9.10 nm), but a normal to elevated number of medium HDL particles (>11 μmol/L). Our data demonstrated that NMR can not only unequivocally detect this lipoprotein abnormality, but also identify the abnormal VLDL accumulation associated with renal disease. Therefore, this rapid procedure may be helpful in the screening and management of this rare disease.

Our study has some limitations. We report analysis of only three cases, but this disease has an extremely low frequency in the general population and, besides, our results seem to be concordant with other studies [7,9,31]. The NMR used by the LipoScale test is not able to quantify small HDL particles, but has been able to identify a distinctive lipoprotein pattern, which allows the clear identification of patients with LCAT deficiency. The first case reported developed renal disease, and, although this could have been caused by other diseases, the patient demonstrated progressive proteinuria, without other pathologies, such as diabetes or hypertension, which could have conditioned the development of nephropathy. Finally, the increase in lipoprotein particles could be the consequence of renal disease, but we had access to the entire clinical records of the third case, and could check that the lipid disorders preceded renal disease.

In conclusion, the current study presents three new cases of LCAT deficiency, which have not been previously described. Two of the three cases are siblings, but only one developed renal disease, although they had the same LCAT genotype and the same LCAT activity, which suggests that renal disease is not only dependent on LCAT deficiency. The classical separation between FED and FLD could be due to the presence of VLDL particles, rich in triglyceride, free cholesterol and LpX, which are possibly responsible for renal disease.

## Figures and Tables

**Figure 1 jcm-08-01860-f001:**
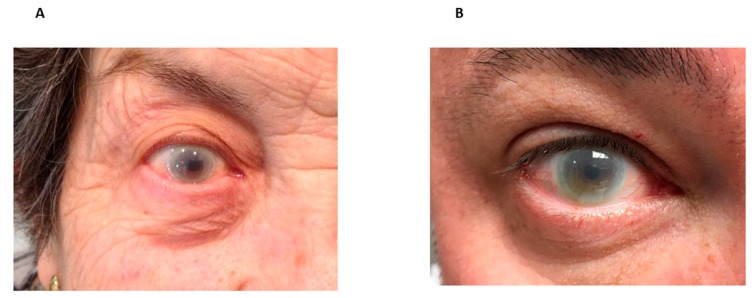
Corneal opacity in FLD patients. Corneal opacity from second case (**A**) and third case (**B**) are shown.

**Figure 2 jcm-08-01860-f002:**
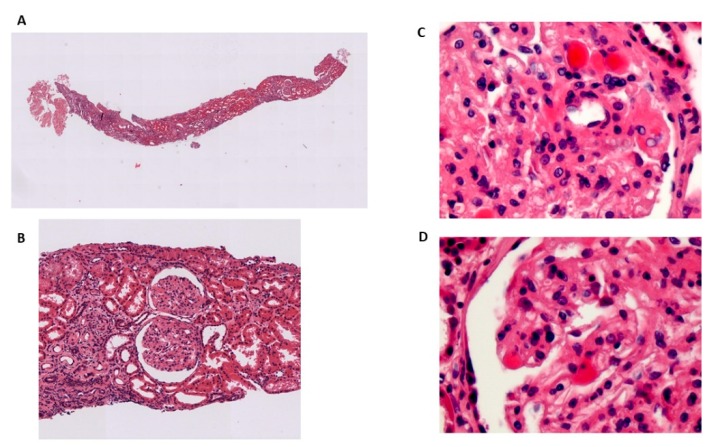
Histological analysis of renal biopsy. (**A**) shows panoramic of the renal tissue cylinder studied; (**B**) shows detail of glomerular involvement, with expansion of the mesangium; (**C** and **D**) show details of thickening of capillary walls, and presence of foamy lipid deposits at capillary and mesangial level.

**Figure 3 jcm-08-01860-f003:**
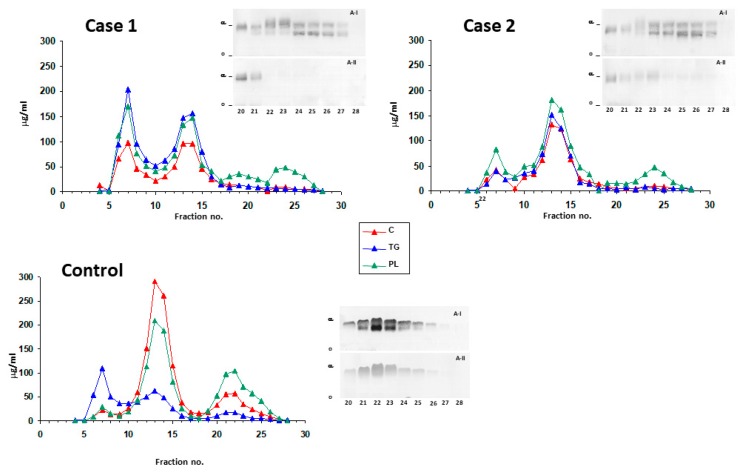
Plasma Gel Filtration Fast Protein Liquid Chromatography (FPLC) analysis. Plasmas were subjected to FPLC, as described in Methods, and the concentrations of cholesterol (C), triglyceride (TG) and choline-containing phospholipids (PL) were determined in the indicated fractions. Inserts: the indicated fractions corresponding to HDL were concentrated, and subjected to agarose gel electrophoresis and immunoblotting for apo A-I and apo A-II. The profiles corresponding to the first case (**A**), second case (**B**) and control subjects (**C**) are presented.

**Figure 4 jcm-08-01860-f004:**
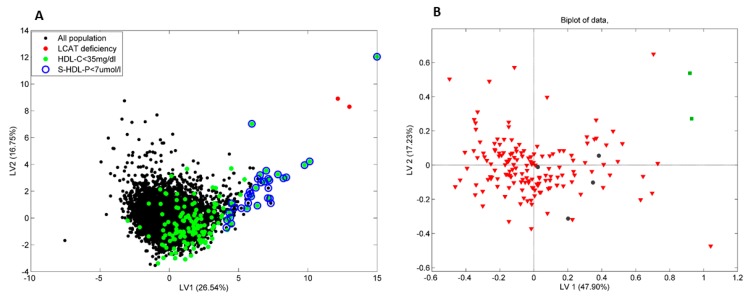
Partial least square discriminant analysis (PLS-DA) models for LCAT deficiency classification, based on NMR Lipoprotein profile data. (**A**) Scores plot from a PLS-DA model designed for LCAT deficiency identification among the general population. The individual projection on the first two latent variables is indicated with a black dot for individuals with HDL-c ≥ 35 and small HDL particle concentration > 7umol/L, with a red dot for LCAT deficiency patients, with a green dot for HDL-C >35 mg/dL, and a blue circle if the small HDL particle concentration is < 7umol/dL. (**B**) Biplot of the PLS-DA model designed for LCAT deficiency identification among patients with very low levels of HDL cholesterol. The individual projection on the first two latent variables is indicated with a red triangle for low HDL-C patients (*n* = 150 individuals), and with a green square for two individuals (2nd and 3rd cases) with LCAT deficiency.

**Figure 5 jcm-08-01860-f005:**
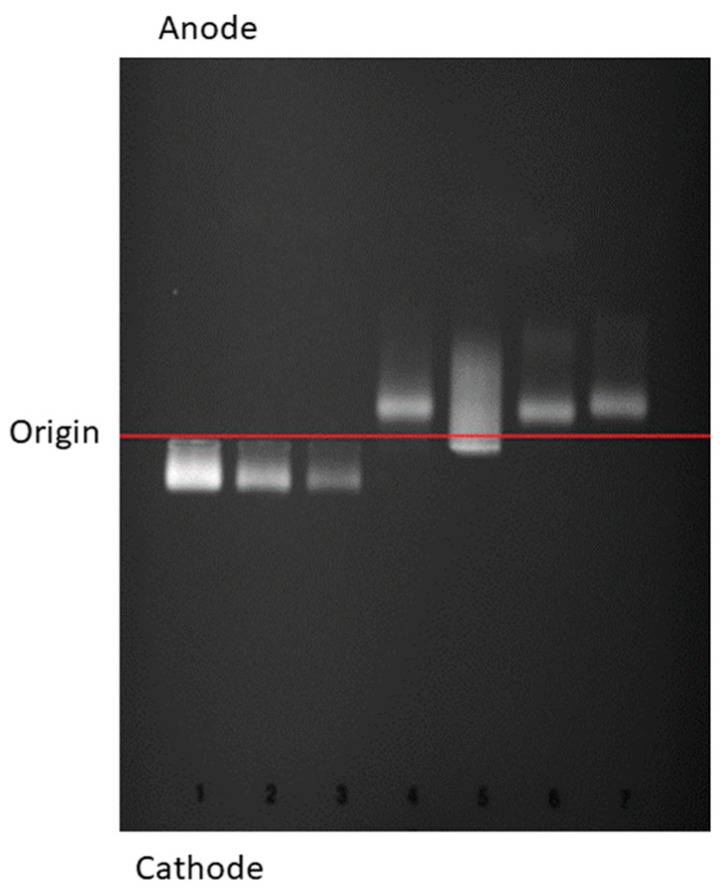
Filipin Staining of synthetic Lp-X and EDTA-plasma after agarose gel electrophoresis. 10 μL of sequential dilutions of synthetic Lp-X were loaded into lanes 1–3, as follows. 1: 0.530 mg/mL cholesterol; 2: 0.265 mg/mL cholesterol; 3: 0.1325 mg/mL cholesterol. 10 μL of fresh EDTA-plasma were loaded into lanes 4–7, as follows: 4, FLD patient 2 (without renal disease); 5, FLD patient 3 (with renal disease); 6 and 7, normolipidemic controls. Origin: application point.

**Table 1 jcm-08-01860-t001:** Lipid profile and lecithin cholesterol acyltransferase (*LCAT)* activity in all cases.

	First Case	Second Case	Third Case	Reference Values
Currently age, years	88	84	39	NA
Age at diagnosis, years	65	78	38	NA
BMI, kg/m^2^	25.3	24.7	24.2	18.5–25
Total cholesterol, mg/dL	119	147	187	150–219
Triglycerides, mg/dL	245	158	181	30–149
Non-HDL cholesterol, mg/dL	112	133	174	100–160
HDL cholesterol mg/dL	7	14	13	40–60
Apoliprotein A1, mg/dL	66	51.3	30	119–155
Apolipoprotein B, mg/dL	96	131	35	73–109
Creatinine, mg/dL	5.35	0.88	2.07	0.51–0.95
Proteinuria, g/L	1.95	Negative	0.77	0.0–0.15
Renal disease	Yes	No	Yes	NA
LCAT genetic analysis	c.605T > C p.(Ile202Thr)	c.605T > C p.(Ile202Thr)	c.512G > A p.(Arg171Glu)	NA
Exogenous LCAT activity (nmol/h*mL)	0.05	1.7	1.35	(23.0–57.0)
Endogenous LCAT activity (Fractional esterification rate, %/h)	1.50	1.40	1.90	>3.50%

BMI: Body Mass Index; NA: Not Applicable.

**Table 2 jcm-08-01860-t002:** Results of Liposcale analysis.

	Large VLDL-P (nmol/L)	Medium VLDL-P (nmol/L)	Small VLDL-P (nmol/L)	Large LDL-P (nmol/L)	Medium LDL-P (nmol/L)	Small LDL-P (nmol/L)	Large HDL-P (μmol/L)	Medium HDL-P (μmol/L)	Small HDL-P (μmol/L)	VLDL-C (mg/dL)	LDL-C (mg/dL)	HDL-C (mg/dL)	VLDL-TG (mg/dL)	LDL-TG (mg/dL)	HDL-TG (mg/dL)
Second case	0.73	3.83	22.6	62.3	407	606.17	0.33	8.35	0.01	7.01	107.44	5.97	40.19	44.43	27.95
Third case	1.60	10.6	55.3	51.1	53.6	145.91	0.25	8.11	0.38	15.20	26.25	22.61	108.23	13.39	11.69
Control population	0.69 ± 0.23	2.61 ± 0.67	20.1 ± 5.45	81.7 ± 19.4	183 ± 65.8	346 ± 78.4	0.24 ± 0.06	8.76 ± 1.97	24.4 ± 7.38	9.71 ± 4.04	81.5 ± 22.9	59.8 ± 15.1	29.2 ± 7.02	11.2 ± 2.73	16.3 ± 7.61

VLDL-P: Very Low-density lipoprotein particles; LDL-P: Low-density lipoprotein particles; HDL-P: High-density lipoprotein particles; VLDL-C: Very Low-density lipoprotein cholesterol; LDL-C: Low-density lipoprotein cholesterol; HDL-C: High-density lipoprotein cholesterol; VLDL-TG: Very low-density lipoprotein-triglyceride; LDL-TG: Low-density lipoprotein-triglyceride; HDL-TG: High-density lipoprotein-triglyceride.

**Table 3 jcm-08-01860-t003:** Percentages of esterified and free cholesterol in the serum of 2 LCAT deficiency samples and 3 controls.

Patient	% Esterified Cholesterol	% Free Cholesterol
Second case	32.9	67.1
Third Case	1.6	98.4
Control	75.3	24.4
Control	67.4	32.6
Control	74.1	25.9

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
