# Peer review of "Lipid Profile Rather Than the LCAT Mutation Explains Renal Disease in Familial LCAT Deficiency"

_jcm, 2019, doi:10.3390/jcm8111860_

Round 1

Reviewer 1 Report

I believe that after revision the manuscript can be published after the following small corrections.

It looks like in phrase (lines 213-214) “… both subjects had  phospholipid and cholesterol-rich VLDL levels compared with the control” word “higher” (after “levels”) was missed.

In the phrase (lines 285-287) “The current study shows that the main cause of mortality in LCAT deficiency does not depend on the responsible LCAT mutation, because two of  our cases are siblings. Sharing the same mutation, just one of them developed renal disease” one word should be added:

“The current study shows that the main cause of mortality in LCAT deficiency does not depend only on the responsible LCAT mutation, because two of our cases are siblings. Sharing the same mutation, just one of them developed renal disease.”

In Supplementary files: names of the files “Supplementa Figure 1” and “Supplementa Figure 2” should be corrected.

Author Response

It looks like in phrase (lines 213-214) “… both subjects had phospholipid and cholesterol-rich VLDL levels compared with the control” word “higher” (after “levels”) was missed.

We agree with the reviewer. According with the reviewer suggestion, we have included the word higher after levels.

In the phrase (lines 285-287) “The current study shows that the main cause of mortality in LCAT deficiency does not depend on the responsible LCAT mutation, because two of our cases are siblings. Sharing the same mutation, just one of them developed renal disease” one word should be added:

“The current study shows that the main cause of mortality in LCAT deficiency does not depend only on the responsible LCAT mutation, because two of our cases are siblings. Sharing the same mutation, just one of them developed renal disease.”

According with the reviewer suggestion, we have included the word “only” in the next sentence: “The current study shows that the main cause of mortality in LCAT deficiency does not depend only on the responsible LCAT mutation, because two of our cases are siblings. Sharing the same mutation, just one of them developed renal disease.”

In Supplementary files: names of the files “Supplementa Figure 1” and “Supplementa Figure 2” should be corrected.

According with the reviewer suggestion, we have corrected the spelling mistake.

Reviewer 2 Report

The authors have much improved the manuscript.

Author Response

The authors have much improved the manuscript

Thank you for your corrections that have helped improve the article

This manuscript is a resubmission of an earlier submission. The following is a list of the peer review reports and author responses from that submission.

Round 1

Reviewer 1 Report

In the manuscript authors describe three FLD patients, with one of them with developed renal disease. Authors discovered never described before two new mutations of LCAT gene.

The knowledge of the mechanisms of development of renal dysfunction in LCAT-deficient patients is significant for development of methods for  treatment and prevention of this disease and obtaining approval for clinical trials of possible new therapies. Prevailing hypothesis associates presence of abnormal lipoprotein LpX in plasma of FLD patients with development by them the renal disease.

Authors studied clinical characteristics, plasma lipid levels, LCAT activity, lipoprotein particle profile by NMR and FPLC, free and esterified cholesterol in three patients. However, their conclusion that “renal disease is not only dependent on LCAT deficiency and could be due to the presence of VLDL particles rich in triglycerides and free cholesterol” is not convincing,  because they did not directly measured LpX in plasma of the patients. Changes in FPLC profiles (Fig. 3) showed that there was increased level of cholesterol and phospholipids in VLDL fractions. However, the size of LpX particles (50-70 nm diameter) is quite similar to the VLDL size (30-80 nm diameter) and it is known that LpX particles are enriched in free cholesterol and phocpolipids (see review Fellin, R. and E. Manzato (2019). "Lipoprotein-X fifty years after its original discovery." Nutrition Metabolism and Cardiovascular Diseases 29(1): 4-8). It is quite possible, that as a result of overlapping of VLDL and LpX particles the observed increased level of cholesterol and phospholipids in VLDL region is the consequence of increased level of LpX.

Authors said that “there are no laboratory methods for direct measurement of LpX”, however such relatively simple method was recently published. It based on agarose gel electrophoresis and staining the gels by filipin, which specifically shows the particles with free cholesterol. See Freeman, L. A., et al. (2019). "Plasma lipoprotein-X quantification on filipin-stained gels: monitoring recombinant LCAT treatment ex vivo." J Lipid Res 60(5): 1050-1057. In the manuscript authors showed the results of agarose gel electrophoresis, but only after staining the gels with Sudan Black. However, this stain is not able to reveal the LpX particles, because it stains only neutral lipids.

Authors provide reference 29, where the mentioned above method for measurement of LpX was successfully used (by the way, in reference 29 the author’s name was missing).

The results of the study permit to suggest that different levels of LpX in plasma of three studied patients are responsible for difference in development of renal disease by them. I recommend the authors to directly measure LpX in plasma of the patients and then return to discussion of the mechanisms by which LCAT deficiency affect the kidney function or change the conclusions in the discussion and abstract and mark that obtained results are consistent with suggestion that LpX plays a crucial role in development of kidney dysfunction in LCAT-deficient patients.

Couples additional comments:

In Introduction describing possible mechanisms of renal disease in LCAT-deficient patients and role of abnormal lipoproteins in their plasma in this pathology authors did not mentioned that these lipoproteins were named LpX. In first phrase of Introduction “Lecithin cholesterol acyltransferase (LCAT) is a lipoprotein-associated enzyme which plays a central role in the esterification of free cholesterol (FC)…” need to add “in plasma”, because in cells esterification of free cholesterol is catalyzed by two other enzymes, namely SOAT1 and SOAT2. In the text of the manuscript authors mentioned that plasma was used for lipoprotein analysis, whereas in abstract it was serum.

Reviewer 2 Report

Interesting and relevant paper. We still do not have treatment for the orphan disease FLD, and description of more cases such as these may help to build registries and work towards actual enzyme of gene therapy.

Solid methods.

Introduction is a bit lengthy (reads like a review); discussion can be shortened and focused as well.

Was LpX detectable by FPLC? Please see PMID:19230892.

Indeed the more severe case seems to have more LpX, this seems supported by the data, and this may have aggrevated his renal disease. Please discuss more specifically and in more mechanistic detail why one FLD patient has more LpX and another FLD patient has less. Could this be dietary? Genetic variants in other lipid regulators? Penetrance of LCAT mutations?